# Early Sport Specialization and Relative Age Effect: Prevalence and Influence on Perceived Competence in Ice Hockey Players

**DOI:** 10.3390/sports10040062

**Published:** 2022-04-18

**Authors:** Vincent Huard Pelletier, Jean Lemoyne

**Affiliations:** 1Department of Human Kinetics, Université du Québec à Trois-Rivières, 3351 Boulevard des Forges, Trois-Rivières, QC G9A 5H7, Canada; jean.lemoyne@uqtr.ca; 2Laboratoire de recherche sur le hockey UQTR, 3351 Boulevard des Forges, Trois-Rivières, QC G9A 5H7, Canada

**Keywords:** relative age effect, early sports specialization, perceived competence, ice hockey, youth sport

## Abstract

The relative age effect (RAE) and early sport specialization (ESS) have been of growing interest in the sports world, especially in ice hockey, because of their potential adverse effects. However, little is known about their distribution within each level of play in Canadian minor ice hockey, or whether they influence young people’s perceived competence, a variable of interest in long-term sports development. A sample of elite adolescent players (N = 204) and a sample of recreational and competitive players (N = 404) were used to measure these constructs, and chi-square tabulations were conducted to compare their distribution. Our results reveal that RAE (χ^2^ = 20.03, *p* < 0.01, Cramer’s V = 0.13) and ESS (χ^2^ = 66.14, *p* < 0.001, Cramer’s V = 0.24) are present, but there are apparently no gender differences in their distributions. Neither the level of RAE nor ESS seems to affect the perceived competence of the players, regardless of gender. The results of this study highlight the presence of RAE and ESS in Canadian minor ice hockey, especially at the elite level, but indicate that they do not affect the self-perception of ice hockey players. Additional research on these concepts is needed to obtain a complete picture of their potential impact on sports development.

## 1. Introduction

The benefits of sustained physical activity on the health of individuals, regardless of age, have long been known and promoted [1]. In Canada, a very popular form of physical activity, especially among the youth, is participation in organized sports. Indeed, 77% of parents reported that their child aged 5 to 19 participated in at least one organized sport, while 66% of young people aged 10 to 15 reported being involved in at least one organized sports activity [2]. Organized sport also has its own set of health benefits. At the physical level, it protects against the chances of suffering from a long-term health problem [3]. We note that young athletes have an aerobic capacity above the 90th percentile, have an advantageous body composition, demonstrate greater strength and enjoy a better general physical condition than their non-sporting peers [4]. The benefits continue at the psychosocial level. Organized sports participation prevents school dropout [5], depressive symptoms and suicidal thoughts [6], while promoting positive self-perceptions and the acquisition of life skills that can be used later in adult life [7]. In Canada, ice hockey offers an interesting example, in that it is the national sport, has a strong governance structure, occupies an important place in popular culture and currently includes over 600,000 players of all ages and profiles [8]. Young people involved in ice hockey, especially at the competitive level, appear to be more physically active than their peers [9] and display signs of good health into adulthood [10]. Unfortunately, organizational factors such as early sports specialization or relative age effect can negatively impact the benefits associated with organized sports participation.

Youth sport has greatly changed over the past 20 years to become a hyper-organized and competitive environment primarily driven by adults [11]. An increasingly prevalent issue in both the research community [12] and the mainstream media is early sports specialization (ESS), a developmental pathway choice generally recognized to have several negative consequences [13]. ESS can be defined using three criteria [14]: (1) intensive participation in organized training or competitive activities over a period of at least 8 months per year; (2) participation in a single sport that interferes with the pursuit of other physical activities (active leisure, other sports); and (3) involvement of prepubescent children or those under the age of 12. Although the disadvantages of this pathway choice are now well known, the fact remains that a large proportion of young ice hockey players continue to engage in it [15]. The reasons for engaging in ESS can be historical, economic or simply due to mistaken beliefs. Papers such as Chase and Simon’s 1973 famous study [16], Bloom’s Developing Talent in Young People [17], or Ericson’s Deliberate Practice Theory [18] have all been used as a justification to encourage the early and repeated practice of an activity to achieve a high level of expertise instead of sport sampling, a much more recommended pathway [14]. Today, the main reason for early specialization is players’ and parents’ belief that it gives them an advantage when seeking to be selected by junior or varsity sports teams or considered for a sports scholarship [19,20]. By definition, ESS significantly increases the chances of falling into overtraining [14]. In addition to promoting premature withdrawal from sports in adolescence, this doubles the risk of suffering an overuse injury [21]. Overuse injuries cause at least 20% of all sport dropouts in elite sport; they are the most important reason for dropout, the first being pressure to perform, another consequence of ESS [22,23]. These are not the only harmful outcomes: young athletes who specialize prematurely risk developing an unhealthy dependency on their coach [12], which predisposes them to abuse. In addition, they experience more stress, are prone to psychological problems [12] and gradually lose their love for physical activity, placing them at greater risk of becoming sedentary adults [24]. The prevalence of specialization has recently been measured in Quebec minor ice hockey [15], but it is not yet known to what extent this phenomenon is present for minors at the highest levels of performance, including national team selections.

Another organizational factor observed in multiple sports for decades is the relative age effect (RAE), which can be defined as the age difference between individuals in the same group [25]. To bring together young people at the same stage of development, ensure that differences in performance are limited and establish a certain level of safety, 1- or 2-year chronological age groups are common in youth sport [26]. Thus, it is possible to end up with two athletes who are 23 months apart playing against each other. This may not seem very much to an adult, but it represents a substantial difference for teenagers. Relative age effect therefore refers to the disadvantages and implications of the interaction between the athlete’s date of birth and the cut-off date used by the sports federation to define age categories [26]. The real problem occurs when this advantage persists over time, offering relatively older players a better chance to participate and develop in the sport, a phenomenon known as the Mathew Effect [27]. For example, a parent may decide not to enroll their child in an organized sport for fear the child would be disadvantaged compared with another who was born earlier and is therefore more physically imposing. With respect to the prevalence of relative age across several game calibers, recent ice hockey data suggest that there is no significant difference between recreational and competitive calibers; however, an RAE is reported in elite selections [28,29]. Hancock states that one reason for the presence of a strong RAE in elite males is that, besides being potentially more imposing physically because of their older age, they may have been exposed to a higher level of competition when young and may therefore possess more quality experience when entering adolescence [30].

As important as RAE or ESS may be in terms of their direct impacts on the development of young athletes, they may also affect another important construct in sports practice: perceived competence. Perceived competence refers to an individual’s perception of their capacities in a certain performance area [31]. Developed by Harter [32], the theory of competence motivation holds that young individuals who consider themselves adept in a specific domain of competence tend to devote more energy to further developing their skills in that area. As a result, skills improvement leads to an increase in perceived competence, resulting in higher autonomous motivation towards an activity [32,33]. RAE has previously been shown to have a slightly negative effect on the perceived competence of young ice hockey players, possibly because relatively younger players are also at a physical disadvantage compared to their peers [28]. ESS also has a negative impact on perceived competence insofar as highly specialized young people tend to compare themselves more to other elite players and receive more frequent corrective and negative feedback from coaches assuming a greater role than they should at this stage of young players’ sports development [34]. Both RAE and ESS, and their influence on perceived sports competence, could theoretically affect the long-term athletic commitment of ice hockey players, whether or not they are involved at the elite level.

This study has two objectives. The first is to measure the prevalence of RAE and ESS within a selection of elite adolescent teams compared with its prevalence in non-elite teams. The second is to establish the effects of RAE and ESS on players’ perceived competence within an elite environment. These objectives are the basis for five hypotheses.

**Hypothesis** **1** **(H1).**
*RAE will be present in higher proportions in the elite selection compared with the non-elite teams.*


**Hypothesis** **2** **(H2).**
*ESS will be present in the same proportions in the elite selection and the non-elite teams.*


**Hypothesis** **3** **(H3).**
*ESS and RAE will be present in the same proportions in both genders.*


**Hypothesis** **4** **(H4).**
*Positive perceived competence will not be present in the same proportion based on level of ESS.*


**Hypothesis** **5** **(H5).**
*Positive perceived competence will not be present in the same proportion based on birth quartiles.*


## 2. Materials and Methods

### 2.1. Sample and Procedures

For this study, 1 primary sample was used to measure the perceived competence, ESS, and RAE of elite ice hockey players, and 2 secondary samples were used to compare these with non-elite ice hockey players. Data for the main sample were collected at a Team Québec selection camp for men (U15) and women (U17) in summer 2021. During this camp, players came to the arena, participated in physical tests and were then evaluated on the ice 2 times a day for 4 days for a chance to make the provincial team. They were asked to complete a questionnaire measuring their perceived competence and ESS during a free period of about 1 h on the second day to ensure they were in a good mental condition. During this collection, 89 males and 115 females were surveyed in compliance with the ethics committee of the researchers’ institution. Players came from all corners of Quebec, all spoke French, and a binary definition of gender was used, as it currently is in the federation. This was the last step in a year-long selection process during which the number of candidates fell from about 204 in the provisional teams to about 45–50 players per gender at the end of this camp. Thus, these players were the best of their cohort, at least according to those involved in the selection process. The second sample had already been used in a research project by the authors of this article. Data for the sample used as a comparison for the prevalence of ESS and RAE in lower-level ice hockey (N = 202 competitive, N = 202 recreative, 15.4 ± 1.9 years old, no female players) were collected in a previous project, and the details surrounding the collection are discussed in this article [9]. In brief, we used two data collection methods during the 2017–2018 hockey season. For the first approach, we identified teams via the website of Quebec’s ice hockey federation, and coaches were contacted to ask for their permission to talk to their players. After explaining the project, the research team issued online or paper questionnaires to those who agreed to participate. For the second approach, we first asked some tournament directors for permission to meet with the participating teams at the registration desk to meet and inform the coaches, who then discussed the project with their players. Research staff members then distributed questionnaires to players who agreed to participate, and they had to be completed during the weekend in their free time, not immediately before or after a game.

### 2.2. Measurement

#### 2.2.1. Early Sports Specialization

The same measurement method for ESS was employed for both data collections [9]. To calculate an ESS score, 3 questions were formulated based on Laprade’s definition, which is the one most frequently used [14]. The first question asked if athletes had participated intensively in ice hockey (development camps, spring league, summer training program) for (1) more than or (2) less than eight months per year. The coded score was 0 for 8 months or less, and 1 for more than 8 months. The second question asked if their ice hockey practice prevented them from participating in other sports activities. The coded score was 0 for no, and 1 for yes. The third question asked for how many years they had been involved intensively in ice hockey. The answer was then deducted from their age to discover if players had this level of commitment before the age of 12. The coded score was 1 for yes, and 0 for no. A 3-point composite score was then created by adding the results of the 3 questions. Three categories of specialization were obtained: <2 = little or no ESS, ≥2 but <3 = moderate ESS and 3 = high ESS.

#### 2.2.2. Relative Age Effect

We measured the prevalence of RAE using the same method as in a previous article [28]. Each birth date was coded into birth quartiles according to the federation cutoff date: Q1 = January–February–March, Q2 = April–May–June, Q3 = July–August–September, and Q4 = October–November–December. We then compared it with the numbers obtained from a national survey for the 2007 year of birth [35].

#### 2.2.3. Ice Hockey Perceived Competence

We measured perceptual competence in ice hockey specifically using a recently validated questionnaire that showed robust psychometrics values (χ^2^_(154)_ = 204.160, *p* = 0.004, RMSEA = 0.021, CFI = 0.997, TLI = 0.993) [36]. The questionnaire consisted of 26 items measured by a 5-point Likert scale and assessed 6 dimensions of ice hockey competence: physical involvement (4 items, ω = 0.95, e.g., “*I win my “one-on-one” battles*”), ice hockey IQ (7 items, ω = 0.97, e.g., “*I take good decision with the puck*”), offensive skills (4 items, ω = 0.95, e.g., “*I have good offensive creativity*”), skating abilities (4 items, ω = 0.95, e.g., “*I am a fast skater*”), resilience (3 items, ω = 0.92, “*I see coaches’ comments as an opportunity to improve*”), and leadership (4 items, ω = 0.94, e.g., “*I stay confident even if playing time is diminished*”). A composite score of all dimensions was then calculated in addition to an overall competence score.

#### 2.2.4. Statistical Analyses

Chi-square cross tabulations were complketed, and Cramer’s V was calculated to assess effect size and interpreted using Volker’s recommendations [37] and in accordance with Delorme [38]. A significant result (*p* < 0.05) indicated a different distribution between the different variables studied, and the interpretation of Cramer’s V was as follows: >0 = very weak, >0.05 = weak, >0.10 = moderate, >0.15 = strong and >0.20 = very strong [39]. We also tested each category with its previous group to confirm if distributions differ from one category to another. Percentage deviation was measured to show the degree to which an observed chi-square cell frequency differs from the value expected on the basis of the null hypothesis.

## 3. Results

Table 1 shows the distribution of RAE by gender and playing level. As can be seen, there is a significant difference in the distribution of RAE by gender (*p* < 0.05) and playing level (*p* < 0.001), with a moderate size effect (V > 0.10) in both cases. These results confirm Hypothesis 1 and partially confirm Hypothesis 3. We also note that the elite players show different distributions regarding RAE; more players are born in Q1 and fewer in Q4 at this playing level compared with other playing levels and the Canadian population of this age.

Table 2 shows the distribution of RAE by gender and playing level. We note there is no significant difference in the distribution of ESS level according to gender (*p* > 0.05). However, a substantial difference in magnitude (*p* < 0.001) exists in ESS levels based on the playing level of the ice hockey players. Specifically, there are fewer competitive players in the high ESS group and more in the low ESS group than at the other two playing levels. Elites also have a significantly larger proportion of highly specialized athletes and a lower proportion of athletes with low ESS. These results refute Hypothesis 2, but partially confirm Hypothesis 3.

Table 3 compares the composite scores for the dimensions of ice hockey competence among elite players. It shows that there is only one significant difference: female players feel less competent in terms of their offensive abilities (*p* < 0.001).

Table 4 represents the distribution of scores regarding the 6 dimensions of perceived ice hockey competence based on SSE and RAE levels. It shows there is no significant difference in perceived competence for these two variables. These results therefore refute Hypotheses 4 and 5.

## 4. Discussion

This study, which was conducted using two samples of adolescent ice hockey players, aimed to describe the prevalence of ESS and RAE at several levels of play and relate them to perceived ice hockey skill, a variable that impacts sports development. We can now review the five hypotheses put forward earlier. First, Hypothesis 1 was confirmed because the distribution of the RAE differed between elite male and female players. It also differed between those playing at other playing levels and in the Canadian population of the same age. The presence of RAE is hardly surprising, as the concept has been part of Canadian ice hockey for over 40 years [25]. Our data are also consistent with the situation in minor Russian ice hockey, where 65% of players were born in the first 6 months of the year [40]. What sets our study apart, however, is that both genders were measured in the elite level and that we also assessed recreative and competitive players. Contrary to the findings of a previous study [28], the proportion of ice hockey players born in the first 2 quartiles of the year was greater in elite players than in the other playing levels, where the RAE was not significantly present. These results make sense to say the very least, because we know that the presence of RAE is strongly influenced by the level of competition. So, it is logical that they are found at a greater proportion in the elite leagues, where players compete fiercely to gain small advantages over their peers and where a difference of 6 or 9 months can have a significant impact. Hypothesis 3 must be partially confirmed regarding the presence of an RAE in both genders, as no significant differences were found within the sample of elite players participating in the national team camps. That ice hockey is Canada’s national sport and is popular with both genders most likely raises the level of competition, even in the female sample, which may well have had less RAE based on previous work [25]. This finding is positive in that it confirms that women’s field hockey has become very competitive in Canada, but is negative in that we are starting to see the same problems as in men’s ice hockey.

Next, we disproved Hypothesis 2, which predicted that the proportion of specialized players would be the same, regardless of playing level. However, the opposite was observed: each playing level had a different ESS distribution than the others. Compared with competitive and recreational players, elite players seem to have a significantly smaller proportion of low or no specialized players and a larger proportion of highly specialized players. Competitive players have many more non-specialized or less specialized players and fewer highly specialized players than other playing levels. At the recreational level, patterns of specialization fall between those observed at the elite and competitive levels. As Quebec ice hockey already demonstrated [15], competitive players show healthier specialization tendencies than recreational players, which can be explained in two ways as follows: (1) these players are usually coached by qualified coaches as opposed to volunteers who are generally involved in recreational play, and (2) they may have better athletic abilities than those playing at a lower level, which predisposes them to a wider range of activities. Elite players, on the other hand, have by far the largest proportion of highly specialized players. This may appear surprising, as they are coached at least as well as athletes at the competitive level and are expected to be even more athletically gifted and well rounded. The reason for their high level of early specialization may be that young players and their parents believe in the importance of being labeled “talented” at a young age in order to gain access to better teams, receive extensive training and compete at a higher level in preparation for an eventual athletic career [12]. Another argument maintains that more specialized young people tend to receive athletic scholarships more often at the college level, as Hockett indicates [19]. While the physical and psychological benefits of sports sampling are well-known, the short-term benefit of ESS can be compelling to many, especially in a major camp such as the national team. The fact that the camp includes 14-year-old ice hockey players may also accentuate this phenomenon, as the selection occurs early in the sport’s development process compared with the junior (15–16) and professional (17–18) drafts.

Regarding SSE, this study’s results partly confirm Hypothesis 3 as well, because no significant difference was found between male and female players. Previous work suggests that girls are more prone to experience SSE, but this may be largely because most studies deal with early developing individual sports such as synchronized swimming or gymnastics [41]. The percentage of highly specialized girls is lower in team sports such as ice hockey; the results of this study therefore support this trend.

The last two hypotheses addressed perceived competence in ice hockey and its relationship with the two constructs under study, RAE and ESS. Contrary to what one might have expected from the literature, the proportion of field hockey players with a positive perception of their skills did not change according to ESS or RAE levels. Indeed, previous work established a link, albeit a relatively weak one, between perception of endurance and ESS [15], while another paper linked perception of physical strength and RAE [28]. In both cases, a broader physical self-perception tool was used, highlighting the importance of using an accurate, sport-specific measure. Alternatively, the literature also suggests that ESS may interfere with the normal development of young athletes by isolating them from their peers, causing overly close bonds with their coaches, and placing them at risk for overtraining [14]. Thanks to the present study, we believe that in the context of Quebec elite ice hockey, even highly specialized players evaluate themselves positively in terms of their skills. This may be because ice hockey is a sport with approximately 20 players per team, where the cost of ice rental makes it more difficult to practice in isolation. Less isolation and the esteem-enhancing qualities of a healthy team environment are conducive to very positive self-perception, which is to be expected from a group of elite players [42,43]. These findings moderate the associations between RAE, SSE and perceived competence, which may be somewhat reassuring for the psychological health of elites who choose to invest in a single sport at a young age or for those disadvantaged by their birthdate.

A strength of this work is the number of concerns it addresses relative to ESS and RAE research. Although these issues have attracted increasing attention in the last 20 years, there is still an absence of data regarding their representation in a particular team sport such as ice hockey at all youth playing levels [13,15]. Furthermore, this study is in line with a recent consensus statement on ESS that highlights the need for conducting research to differentiate gender-based patterns, limit the use of adult samples to reduce the possible drawbacks of a multiple-year recall, and further explore the impacts of ESS [44]. A second strength of our study is that it shows that RAE and ESS have apparently no impact on the perceived competence of ice hockey players. This improves our understanding of these issues, but also raises specific questions concerning ice hockey. First, might the relationship with perceived competence be different if it were measured at ages 18, 20, and 25? Do players with low ESS playing at competitive levels tend to become more successful later in their development, or does the advantage that those with high ESS gain in early adolescence carry over into adulthood? Although both of these questions could be answered using a longitudinal research design, such was unfortunately not possible in this project. Additionally, a more precise operational definition of ESS could better categorize young athletes’ level of specialization. LaPrade’s current three-indicator definition is slightly too broad and does not necessarily point to potentially negative behaviors and variables about training load [14]. For example, is it safe to say that a 10-year-old playing ice hockey for 10 months and enrolling in soccer for 2 months while continuing to train for ice hockey (non-specialized by the current definition) is at less risk of adverse psychological or physical consequences than a player who does not enroll in soccer, but has the same practice in ice hockey (specialized by the current description)? Additional data regarding active behaviors and the physical and psychological impacts of SSE could be added to better identify youth with a genuinely problematic sports development [44].

RAE has been present for at least a hundred years in practically all sports, and it affects both men and women at an equivalent level of competition [45]. This is a difficult phenomenon to work around logistically, since separating players by birthdates and having them play together in age-based groups of 24 months as in ice hockey is a simple, safe and inexpensive solution that ensures a relatively homogeneous performance level [25]. However, a very high RAE between the Canadian population and elite level ice hockey players may be concerning, especially because there is a RAE between the elites and other levels of play as well. The visibility and access to qualified coaches these players enjoy in comparison to their peers confer a distinct advantage. Thus, initiatives such as tournaments reserved for the best players born in Q3 and Q4 could partially reduce the RAE by offering these neglected players additional opportunities to showcase their skills. At the very least, we may discover that players disadvantaged by their later birthdate do not have a lower perceived competence than others. This opens the door to research on other psychological variables of interest, including perceived enjoyment, stress, burnout, or social support, to reach solid conclusions about the real impact of RAE and ESS on youth sports development [46].

## 5. Conclusions

In conclusion, this study explores the distribution of the relative age effect as well as early sports specialization within Canadian minor ice hockey in function of the different playing levels and the two genders. It confirms that these two issues are present in Canadian elite ice hockey, especially at the elite level, but do not affect players’ perception of competence. Although this research adds to the present ESS and RAE literature, it is purely descriptive and somewhat limited by its cross-sectional design. Further research is needed to investigate the additional impacts they may have on the athletic development and long-term physical activity of these young athletes. Awareness of the harmful effects of ESS and RAE must continue in youth sport, especially at the elite level.

## Figures and Tables

**Table 1 sports-10-00062-t001:** RAE prevalence across playing level and gender.

Birth Quartile	Canadian Birth Percentage (2007)	Gender N (Percentage Variation)		χ^2^	Playing Level N (Percentage Variation)			χ^2^
	Both Genders *	Male	Female		Recreative ***	Competitive	Elite **	
Q1	23	38 (−6.7)	56 (+5.2)	2.46 (*p* = 0.48) Cramer’s V = 0.11	64 (−9.3)	58 (−19.8)	94 (+28.6)	20.03 (*p* < 0.01) Cramer’s V = 0.13
Q2	25	23 (+17.9)	22 (−13.7)		49 (−3.2)	61 (+17.5)	45 (−14.2)	
Q3	27	20 (+7.3)	23 (−5.6)		44 (+5.2)	41 (−4.4)	43 (−0.7)	
Q4	25	7 (−23.1)	14 (+17.7)		39 (+18.2)	41 (+21.2)	21 (−38.5)	

* There is a significant difference (*p* < 0.01) between Canadian birth and male (*p* < 0.01, Cramer’s V = 0.28) and female (*p* < 0.01, Cramer’s V = 0.28) elite players. ** Elite is significantly different from recreative (*p* = 0.01, Cramer’s V = 0.16) and competitive (*p* < 0.001, Cramer’s V = 0.21). *** The competitive and recreative samples contain only male participants.

**Table 2 sports-10-00062-t002:** ESS prevalence across playing level and gender.

ESS Score	Gender N (% Variation)		χ^2^	Playing Level N (% Variation)			χ^2^
	Male	Female		Recreative ***	Competitive *	Elite **	
Low	3 (−23.3)	8 (+16.5)	χ^2^ = 0.98 Cramer’s V = 0.07	40 (−5.5)	73 (+72.5)	11 (−72.1)	χ^2^ = 66.14 * Cramer’s V = 0.24
Moderate	27 (+1.4)	36 (−1.0)		61 (−8.3)	71 (+6.7)	63 (+1.7)	
High	48 (+1.5)	66 (−1.1)		101 (+8.4)	58 (−37.7)	114 (+31.5)	

* Competitive differs significantly from recreative (*p* < 0.001, Cramer’s V = 0.23) and elite (*p* < 0.001, Cramer’s V = 0.41). ** Elite differs significantly from recreative (*p* < 0.001, Cramer’s V = 0.21). *** The competitive and recreative samples contain only male participants.

**Table 3 sports-10-00062-t003:** RAE association with ice hockey perceived competence.

Construct	Mean Scores	
	Male	Female
Perceived offensive abilities	4.30	4.15 **
Perceived skating abilities	4.27	4.39
Perceived physical imnvolvement	4.25	4.16
Perceived hockey sense	4.29	4.21
Perceived ability to lead	4.47	4.50
Perceived ability to face adversity	4.22	4.23

** Significant difference at 0.001.

**Table 4 sports-10-00062-t004:** ESS and RAE association with ice hockey perceived competence.

Perceived Competence	ESS				χ^2^	RAE				χ^2^
	Scores	Low	Moderate	High		Q1	Q2	Q3	Q4	
Offensive abilities	3.0	0	0	0	χ^2^ = 6.546 Cramer’s V = 0.140	0	0	0	0	χ^2^ = 6.189 Cramer’s V = 0.110
	3.5	0	5	7		7	1	3	1	
	4.0	1	13	32		23	12	7	6	
	4.5	7	23	32		31	17	8	6	
	5.0	3	17	28		21	10	13	5	
Defensive abilities	3.0	0	0	4	χ^2^ = 9.170 Cramer’s V = 0.164	1	1	0	4	χ^2^ = 11.605 Cramer’s V = 0.150
	3.5	0	2	8		6	0	2	10	
	4.0	1	13	26		22	7	3	42	
	4.5	3	19	21		19	12	7	44	
	5.0	7	24	42		34	12	7	73	
Physical involvement	3.0	0	0	1	χ^2^ = 8.487 Cramer’s V = 0.59	0	1	0	0	χ^2^ = 13.168 Cramer’s V = 0.160
	3.5	0	2	11		7	3	2	1	
	4.0	5	16	34		23	13	16	4	
	4.5	4	17	30		30	9	8	6	
	5.0	2	22	24		22	14	5	7	
Hockey sense	3.0	0	4	5	χ^2^ = 4.777 Cramer’s V = 0.119	5	2	2	0	χ^2^ = 6.762 Cramer’s V = 0.114
	3.5	0	5	9		9	3	3	0	
	4.0	3	16	24		19	11	10	4	
	4.5	2	15	32		25	10	9	6	
	5.0	6	18	31		24	14	8	9	
Ability to lead	3.0	0	0	0	χ^2^ = 4.135 Cramer’s V = 0.111	0	0	0	0	χ^2^ = 8.818 Cramer’s V = 0.131
	3.5	0	2	8		4	1	4	1	
	4.0	2	8	18		10	8	7	3	
	4.5	2	20	27		24	13	9	3	
	50	7	27	48		44	17	12	12	
Ability to face adversity	3.0	0	2	8	χ^2^ = 4.846 Cramer’s V = 0.119	3	2	5	0	χ^2^ = 15.557 Cramer’s V = 0.173
	3.5	0	4	8		7	1	5	0	
	4.0	3	18	31		24	14	9	6	
	4.5	2	10	21		17	9	4	4	
	5.0	6	24	33		31	14	9	9	

## Data Availability

The authors are committed to providing all data used to conduct this study upon request.

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
