# Peer review of "Early Sport Specialization and Relative Age Effect: Prevalence and Influence on Perceived Competence in Ice Hockey Players"

_sports, 2022, doi:10.3390/sports10040062_

Round 1

Reviewer 1 Report

This is a well written and well researched paper, examining important issues relating to RAE, ESS, and perceptions of competence, in the context of Canadian ice hockey. Based on appropriately collected and analysed data, the paper provides a series of findings and recommendations that make a contribution to the literature. The paper is also well aligned with the focus of the special issue.

I have only a few minor comments/suggestions for consideration:

Line 56: Check spelling of Ericsson, and in reference list

Line 57: what is the incorrect interpretation being referred to?  And what is the correct interpretation?  Add references to these points, if possible.

Line 140: In relation to the comment 'best of their cohort', consider whether need to add a qualifier along lines of: 'at least according to those involved in the selection process.'

Discussion of gender in paper is organised around understanding of gender in binary terms ie both genders.  Given more recent understandings of gender as exising on a spectrum, consider adding some further context eg (if this is the case)-in this paper, gender is used in a binary mode as that is how ice hockey is organised in Canada, but it is recognised that understandings of gender are changing.

Line 232: may be a word(s) missing between the words 'players' and 'those'

Author Response

Thank you for our usefull comments. You can see our replies in the Word file attached.

Reviewer 2 Report

Dear authors, I congratulate your effort and the results obtained in this manuscript. I consider the document correctly prepared, but I will point out some elements that may help to improve your manuscript, with a view to publication.

The introduction seems correct and sufficiently supported by relatively recent literature.

The methods must be clearly described and not refer to one or more articles. This relationship can and should be maintained, but it does not dispense with the explanation of the methodological options.

The authors refer to the use of a recently validated questionnaire. What is the name of the questionnaire? The internal reliability coefficient follows that McDonald's Omega? What are the fit indices obtained in the original validation? And in your work? Due to the size of the sample, they are in a position to prepare a CFA. For each dimension of the questionnaire, they must also present a type question.

The results are clear.

We are of the opinion that the discussion of the results should be contained in a section and the conclusion in another section, which would contain a clear answer to the study objective, limitations, future research perspectives and naturally concrete practical applications of your results to the real context.

Best wishes for a good work.

Author Response

Thank you for your insightful comments that helped raise the level of this study. You can see our responses in the attached Word document.

Reviewer 3 Report

Dear Authors,

The manuscript was conducted using 2 samples of adolescent ice hockey players, it aimed to describe the prevalence of relative age effect (RAE) and early sport specialization (ESS) in several levels of play, and relate them to perceived ice hockey skills, and whether they influence young people's perceived competence. Results of the work are well presented and can be of interest for stakeholders.

However, 2 secondary samples were used as control groups, but their descriptions are not reported, they are only referred to as collected in a previous project. A detailed description of these secondary samples would make the protocol and results presentation easier to follow.

Point by point comments for the authors:

Lines 24-43: in the first part of the introduction some of the text appear to be in bold font.

Lines 127-143: a detailed description of the 2 secondary samples would make the results interpretation easier to follow.

Line 142: it appears that 1 of the secondary samples is an all-male group, this should be pointed out when results are presented.

Lines 231-233: “First, hypothesis 1 was confirmed because the distribution of the RAE differed between elite male and female players those at other playing levels and in the Canadian population of the same age. The sentence is hard to read, maybe a comma between players and those could make it clearer.

Lines 241-242: “These results make sense to say the very least, because we know the presence of RAE is strongly influenced by the level of competition, which is higher for elite players.” This sentence could be rewritten to better clarify the concept significance.

Lines 281-282: gymnastics is listed twice.

Lines 294-296: “Thanks to the present study, we believe that in the context of Quebec elite ice hockey, even highly specialized players evaluate themselves positively in terms of their skills.” This is an interesting result; you could corroborate your discussion by adding some references about the topic of less isolation conducive to positive self-perception in team sports.

Line 319: here a reference to Jayanthi's work is needed. Also, an explanation on why its definition is a bit broad, would better clarify your line of reasoning.

Author Response

Thank you for your insightful comments that helped raise the level of this study, it is much appreciated.  You can see our responses in the attached Word document.
